# Galectin-1 in Pancreatic Ductal Adenocarcinoma: Bridging Tumor Biology, Immune Evasion, and Therapeutic Opportunities

**DOI:** 10.3390/ijms242115500

**Published:** 2023-10-24

**Authors:** Ana Bogut, Bojan Stojanovic, Marina Jovanovic, Milica Dimitrijevic Stojanovic, Nevena Gajovic, Bojana S. Stojanovic, Goran Balovic, Milan Jovanovic, Aleksandar Lazovic, Milos Mirovic, Milena Jurisevic, Ivan Jovanovic, Violeta Mladenovic

**Affiliations:** 1City Medical Emergency Department, 11000 Belgrade, Serbia; bogutova12@gmail.com; 2Department of Surgery, Faculty of Medical Sciences, University of Kragujevac, 34000 Kragujevac, Serbia; bojan.stojanovic01@gmail.com (B.S.); gbalovic@gmail.com (G.B.); 3Department of General Surgery, University Clinical Center Kragujevac, 34000 Kragujevac, Serbia; royalmilnikstaff@gmail.com; 4Department of Internal Medicine, Faculty of Medical Sciences, University of Kragujevac, 34000 Kragujevac, Serbia; marinna034@gmail.com (M.J.); vikicam2004@gmail.com (V.M.); 5Department of Pathology, Faculty of Medical Sciences, University of Kragujevac, 34000 Kragujevac, Serbia; milicadimitrijevic@yahoo.com; 6Center for Molecular Medicine and Stem Cell Research, Faculty of Medical Sciences, University of Kragujevac, 34000 Kragujevac, Serbia; ivanjovanovic77@gmail.com; 7Department of Pathophysiology, Faculty of Medical Sciences, University of Kragujevac, 34000 Kragujevac, Serbia; bojana.stojanovic04@gmail.com; 8Department of Abdominal Surgery, Military Medical Academy, 11000 Belgrade, Serbia; milan.jovanovic@vma.mod.gov.rs; 9Department of Surgery, General Hospital of Kotor, 85330 Kotor, Montenegro; mirovic.milos91@gmail.com; 10Department of Clinical Pharmacy, Faculty of Medical Sciences, University of Kragujevac, 34000 Kragujevac, Serbia

**Keywords:** pancreatic ductal adenocarcinoma, Galectin-1, tumor microenvironment, angiogenesis, immune modulation

## Abstract

Pancreatic Ductal Adenocarcinoma (PDAC) remains one of the most challenging malignancies to treat, with a complex interplay of molecular pathways contributing to its aggressive nature. Galectin-1 (Gal-1), a member of the galectin family, has emerged as a pivotal player in the PDAC microenvironment, influencing various aspects from tumor growth and angiogenesis to immune modulation. This review provides a comprehensive overview of the multifaceted role of Galectin-1 in PDAC. We delve into its contributions to tumor stroma remodeling, angiogenesis, metabolic reprogramming, and potential implications for therapeutic interventions. The challenges associated with targeting Gal-1 are discussed, given its pleiotropic functions and complexities in different cellular conditions. Additionally, the promising prospects of Gal-1 inhibition, including the utilization of nanotechnology and theranostics, are highlighted. By integrating recent findings and shedding light on the intricacies of Gal-1’s involvement in PDAC, this review aims to provide insights that could guide future research and therapeutic strategies.

## 1. Introduction

Pancreatic ductal adenocarcinoma (PDAC), constituting approximately 85% to 95% of all pancreatic cancer cases, is notoriously aggressive, ranking as the fourth leading cause of cancer-related deaths worldwide [1]. The 5-year survival rate fluctuates between 5% and 20% depending on the stage and treatment, with a median survival of merely 4 to 6 months in resistant cases [2,3]. Factors contributing to this grim prognosis include late-stage diagnosis, rapid metastasis, extensive desmoplasia in the tumor microenvironment, and limited efficacy of conventional treatments [3,4,5].

Despite surgical interventions and advances in therapeutic approaches, long-term survival (≥5 years) remains rare in PDAC, and even very long-term survival (≥10 years) is exceptional [6]. Variations in survival rates may suggest an underlying molecular phenotype affecting outcomes [4]. Additionally, the resistance of PDAC to therapies is enhanced by a hostile tumor microenvironment [7,8]. It is anticipated to become the second leading cause of cancer-related deaths in the United States by 2030 [9]. The increasing incidence of PDAC has been associated with various risk factors, including smoking, obesity, diabetes, and chronic pancreatitis. This underscores the urgent need for improved early detection and treatment strategies [10,11]. Ongoing research is critical to enhancing our comprehension of this complex disease, with a particular focus on molecular mechanisms and biomarkers that may guide personalized treatment and foster more effective therapeutic approaches for all patients [12].

Galectins form a sophisticated family of proteins recognized for their pronounced affinity to bind β-galactosides through their carbohydrate-recognition domains (CRDs) [13]. Divided into three principal subtypes—prototype galectins, tandem repeat galectins, and the unique chimeric galectin—each subtype presents distinguishing structural characteristics [13,14]. Prototype galectins include members such as galectin-1, -2, -5, -7, -10, -11, -13, -14, and -15, each with a singular CRD that can form homodimers [15]. In contrast, tandem repeat galectins, including galectin-4, -6, -8, -9, and -12, have two CRDs conjoined by a linker, while the chimeric galectin-3 is marked by a CRD integrated with proline- and glycine-rich sequences [14]. Of particular interest is Gal-1, a member that plays a pivotal role in a myriad of biological processes from modulating the immune system to influencing cancer pathogenesis [16]. By binding to glycoconjugates containing β-galactose, particularly glycans bearing the N-acetyllactosamine (LacNAc) structure, galectins interact with diverse cellular elements [17]. This binding specificity among galectin members steers a range of cellular functions, including proliferation, migration, adhesion, apoptosis, inflammation, and carcinogenesis [18,19].

In the complex landscape of PDAC, Gal-1 and Gal-3 emerge as pivotal areas of research interest. [19,20]. Recognizing its prominence within the tumor microenvironment of PDAC, scientists have highlighted Gal-1 as a potential diagnostic biomarker, prognostic indicator, and therapeutic target [19,21,22]. The ensuing sections of this review are dedicated to unraveling the nuanced relationships between Gal-1 and PDAC. We will explore its structural attributes, its significance in the clinical context of PDAC, and the possibilities and challenges posed by targeting Gal-1 in therapeutic strategies. By delving into the role of Gal-1 within PDAC, we aim to enhance the current understanding and pave the way for more effective strategies against this aggressive malignancy. Additionally, Table 1 offers a clear and organized overview of the role and impact of Gal-1 in PDAC, encapsulating its critical interactions and implications in the progression of the disease.

## 2. Stromal Complexity and Immune Landscape in Pancreatic Ductal Adenocarcinoma: Challenges and Therapeutic Perspectives

Pancreatic ductal adenocarcinoma is characterized by its intricate stromal microenvironment and a complex immune landscape [23]. The stromal components of PDAC, which make up a substantial part of the tumor’s size, play a role in the complexities associated with treating this aggressive cancer [24]. The dense stromal desmoplasia, driven by the interaction between activated Pancreatic Stellate Cells (PSCs) and cancer cells, fuels malignant behavior, while the immunosuppressive nature of the PDAC microenvironment hinders the development of effective immunotherapies [25,26]. This section will explore the critical interplay between tumor cells and stromal components, focusing on both the desmoplastic reaction and the immune landscape. An in-depth understanding of these aspects is essential for paving the way toward innovative interventions that can disrupt tumor-promoting mechanisms and enhance patient outcomes.

### 2.1. Stromal Desmoplastic Reaction and Pancreatic Stellate Cells in Pancreatic Ductal Adenocarcinoma

The stromal microenvironment in PDAC comprises up to 90% of the tumor volume and is composed of extracellular matrix proteins, immune cells, endothelial cells, and cancer-associated fibroblasts [24]. PDAC is characterized by extensive tumor–stroma crosstalk that promotes tumor progression [27]. Targeting the stroma has emerged as a promising approach, but the complex interplay between tumor cells and stromal components requires a nuanced understanding for the development of effective treatment strategies [24].

PDAC is marked by a prominent stromal desmoplasia, consisting of a dense extracellular matrix that obstructs chemotherapeutic drug delivery, facilitating the malignant behavior of cancer cells [7]. This desmoplastic reaction creates a hypoxic environment that supports tumor growth, invasion, and metastasis while resisting conventional therapies [4]. Emerging evidence suggests that targeting PDAC-associated desmoplasia holds significant promise as a novel therapeutic approach for pancreatic cancer treatment [28].

Pancreatic stellate cells are specialized cells residing in the pancreas, primarily known for their role in fibrosis during pancreatic injury. In the context of pancreatic ductal adenocarcinoma, PSCs have been identified as principal contributors to the desmoplastic response [25]. The desmoplastic reaction, characterized by the abundant production of the extracellular matrix and the formation of a fibrous stroma around the tumor, creates a conducive environment for tumor progression [29]. It is in this stromal microenvironment that PSCs play a crucial role. When activated, these stellate cells undergo morphological and functional changes that lead to the secretion of various cytokines, growth factors, and extracellular matrix proteins [27]. These secretions are of particular concern as they fuel malignant tendencies, such as rapid tumor cell multiplication and both local and distant metastasis [27]. Additionally, the interaction between activated PSCs and pancreatic cancer cells is crucial in promoting the progression of the disease [25]. This interaction is largely mediated by paracrine signaling, where cells communicate by releasing factors that affect nearby cells. Research underscores the significance of this paracrine signaling between PSCs and cancer cells [30,31]. Understanding this interplay and its role in disease progression suggests that disrupting this interaction could offer a promising therapeutic strategy for advanced PDAC. The intricate involvement of PSCs as vital components of the stroma microenvironment and as significant contributors to the ECM highlights the need for a deeper exploration of their role in pancreatic cancer.

### 2.2. Immune Landscape and Therapeutic Challenges in Pancreatic Ductal Adenocarcinoma

In pancreatic ductal adenocarcinoma, the tumor stroma’s immune component is crucial, shaping an immunosuppressive microenvironment that hinders effective therapeutic interventions [32]. In the tumor microenvironment, an immunosuppressive milieu is established, largely facilitating tumor immune evasion [33]. This suppressive environment is populated by a diverse array of cells that act in concert to dampen anti-tumor immune responses [26]. Among them, M2 macrophages, a subtype of macrophages, play a critical role. Unlike their M1 counterparts, which exhibit pro-inflammatory and tumor-suppressing activities, M2 macrophages are often referred to as “tumor-associated macrophages” due to their role in promoting tumor growth [34]. They do this by releasing a range of anti-inflammatory cytokines, supporting angiogenesis, and fostering tissue remodeling, all of which can indirectly aid tumor progression [34].

Similarly, regulatory T-lymphocytes (Tregs) are another subset of immune cells that contribute to the immunosuppressive environment [35]. Naturally existing to prevent autoimmune reactions in healthy conditions, Tregs, when present in the tumor microenvironment, can suppress the activity of effector T cells, thereby hindering their ability to target and eliminate tumor cells [35]. Together with myeloid-derived suppressor cells, both M2 macrophages and Tregs synergistically impede T cell activation and obstruct the eliminative functions of cytotoxic T-lymphocytes (CTLs), thereby supporting the tumor’s ability to evade the immune system [4,36,37]. Compounding this challenge, the pancreatic tumor cells themselves produce and release potent immunosuppressive agents, such as Transforming Growth Factor-beta (TGF-β), Interleukin-10 (IL-10), Interleukin-6 (IL-6), Vascular Endothelial Growth Factor (VEGF), and the Fas ligand [38,39]. This complex interplay not only dampens immune surveillance but also diminishes the response to immunotherapies [40]. Although tumor-infiltrating CTLs are present, the PDAC tissue inherently resists CTL infiltration, enabling malignant cells to elude immune detection and conferring them with a survival advantage in this intricate immunological landscape [41]. To overcome these considerable challenges, a deep understanding of the tumor microenvironment and the unique features of pancreatic cancer is crucial [42]. Future interventions must disrupt tumor-promoting mechanisms while maintaining beneficial stromal functions. Developing targeted immunotherapies demands a focus on the intricate mechanisms that allow pancreatic carcinoma cells to evade CTL-mediated eradication [43]. Insight into these aspects could pave the way for innovative therapeutic strategies that enhance anti-tumor immune responses, ultimately improving patient outcomes. For a summarized overview of key features and implications within the PDAC microenvironment, refer to Table 2.

### 2.3. The Role of Immunotherapies in PDAC Treatment

The application of immunotherapy in treating PDAC has been a focal point of research, especially in light of the broader successes achieved with these treatments in other malignancies [44]. While the pioneering work on cancer immunotherapy by Allison JP and Honjo T earned them the 2018 Nobel Prize in Physiology or Medicine, particularly for their advancements with immune checkpoint inhibitors like anti-Cytotoxic T-Lymphocyte-Associated Protein 4 (CTLA-4) and anti-Programmed Death-1 (PD-1) antibodies, the outcomes of such treatments in PDAC have been less promising [45].

Specifically for PDAC, the inherent challenges lie in the tumor’s low intrinsic antigenicity, defective antigen presentation mechanisms, and a dominant immunosuppressive microenvironment characterized by Myeloid-Derived Suppressor Cells (MDSC) and Regulatory T cells (Tregs) [46]. Consequently, while immunotherapies such as checkpoint inhibitors show promise in many cancers, their efficacy in PDAC remains limited, necessitating a deeper understanding and tailored approaches for this particular malignancy [44].

Notably, in 2018, the American Society of Clinical Oncology (ASCO) updated their clinical practice guidelines for metastatic pancreatic cancer, indicating that the PD-1 immune checkpoint inhibitor pembrolizumab is recommended as second-line therapy, but only for those patients who test positive for mismatch repair deficiency or MSI h [47]. This recommendation underscores the importance of routine testing for MSI h in PDAC patients who might be candidates for checkpoint inhibitor therapy, even though the incidence of MSI h in PDAC patients is notably low [47].

Among the types of immunotherapies, anti-CTLA-4 antibodies, like ipilimumab, have shown effectiveness in treating melanoma and renal cell carcinoma. Yet, their application in PDAC remains limited [48]. Adoptive T cell transfer therapy, particularly CAR T cell immunotherapy, has shown promising results in blood malignancies [49]. CAR T cells are designed to target specific tumor antigens, yet their efficacy in advanced PDAC remains underwhelming [50].

Lastly, vaccine-based immunotherapies aim to bolster a patient’s tumor-specific immunity [51]. While the potential of this approach remains to be fully realized in PDAC, the advancements in technology, like next-generation sequencing (NGS), are ushering in new possibilities for personalized vaccine development [47].

In conclusion, while immunotherapies hold immense potential in the realm of cancer treatment, their application in PDAC is still evolving, demanding further research to unlock their full therapeutic potential.

## 3. Galectin-1: Structural Complexity, Biological Function, and Emerging Role in Cancer Treatment

Galectin-1 is a unique and critical member of the galectin family, characterized by its ability to bind to β-galactoside-containing glycoconjugates [52]. Existing as a homodimer with 14-kDa subunits, it can be found in various tissues and demonstrates different subcellular localizations, such as in the cell nucleus, cytoplasm, cell surface, and extracellular matrix [13,53]. Gal-1 plays significant roles in normal and pathological conditions, being involved in diverse cellular processes such as cell cycle control, migration, adhesion, immune responses, apoptosis, and inflammation [54]. Its complex interactions can be sugar-dependent or sugar-independent and may be inhibited by specific compounds like thiodigalactoside (TDG) or lactose [54].

The molecular structure of Gal-1, encoded by the LGALS1 gene at 22q13.1, features a complex β-sandwich composition with two antiparallel β-sheets formed by five (F1-F5) and six (S1-S6a/b) strands, respectively [21]. In humans, it exists as a dimer stabilized by hydrophobic interactions within its core, formed by key residues from both subunits [52]. Additionally, a well-defined hydrogen bond network is established among specific residues [55]. Notably, the presence of six cysteine residues in the Gal-1 sequence makes it susceptible to oxidation, which might lead to changes in its physiological activity and function [56]. Refer to Figure 1 for a visual representation of the Gal-1 molecular structure and its various cellular localizations, underscoring its diverse and crucial roles in cellular mechanisms.

### Galectin-1 in Oncology: From Tumor Progression to Therapeutic Potential

Galectin-1 consistently manifests heightened expression within tumor tissues, encompassing not only the neoplastic cells but also the accompanying stromal cells [16]. Notably, it plays a significant role in the carcinogenesis of colorectal tumors [57,58,59]. Such heightened levels are often indicative of a more aggressive tumor behavior, promoting the development of a metastatic phenotype [60]. Beyond its association with tumor malignancy, Gal-1 exerts a significant influence in fostering tumor angiogenesis through modulation of endothelial cell activity, and it enhances the ability of tumor cells to evade immune surveillance [21,61]. Intriguingly, beyond its role in malignancies, Gal-1 has a contributory role in other inflammatory conditions of the gastrointestinal tract and in diseases that have recently gained significant attention, such as COVID-19 and ulcerative colitis [62,63].

Gal-1’s prominence in a variety of human cancers underscores its pivotal role in oncology [60]. Its diverse interactions in cancer, from nurturing metastatic tendencies to promoting tumor angiogenesis and helping tumors evade the immune system, accentuate its integral position in the complex landscape of cancer biology [64,65]. Given its profound influence, targeting Gal-1 might emerge as a promising strategy for therapies across multiple cancer forms [66].

## 4. Galectin-1 Expression in Normal and Pathological Pancreatic Tissues

In normal pancreatic tissues, Gal-1 expression manifests in a nuanced manner [67]. While most regions of the normal pancreas display weak or absent Gal-1 staining, in situ hybridization has identified pronounced Gal-1 mRNA signals in pancreatic nerves [68,69]. Additionally, immunohistochemical analyses have detected subtle Gal-1 immunostaining in select fibroblasts located within the intra- and interlobular stroma of pancreas [68]. The heightened expression of Gal-1 in both nerves and fibroblasts may suggest their potential roles in intercellular communication or tissue repair processes, pointing to a specialized function of Gal-1 within these specific cell types in the pancreas.

In pathological contexts of the pancreas, there is a marked upregulation of Gal-1 expression [70,71]. For instance, the studies of Wang et al. discerned moderate to intense Gal-1 expression predominantly in the fibroblasts of chronic pancreatitis specimens [71]. Furthermore, Pancreatic Stellate Cells/fibroblasts, pivotal in mediating desmoplastic responses across a spectrum of pancreatic diseases—from chronic pancreatitis (CP) and intraductal papillary mucinous neoplasms (IPMN) to pancreatic intraepithelial neoplasia (PanIN) and pancreatic ductal adenocarcinoma (PDAC)—exhibit pronounced Gal-1 expression [22]. Intriguingly, while ductal cells remain devoid of Gal-1, its presence is concentrated within the cytoplasm, nuclei, and extracellular matrix of Pancreatic Stellate Cells [22]. This elevated expression in Pancreatic Stellate Cells underscores Gal-1’s potential role in modulating stromal responses, intercellular interactions, and possibly the progression of pancreatic pathologies.

### 4.1. Galectin-1 Expression in Pancreatic Ductal Adenocarcinoma

The role of Gal-1 in PDAC has garnered significant attention in the research community, shedding light on its specific localization and implications in the pathology of the disease [21]. In PDAC, a distinct concentration of Gal-1 is observed within stromal myofibroblasts, especially within activated PSCs embedded in the tumor-associated stroma [68]. The expression of Gal-1 in these stromal constituents suggests its potential involvement in stromal cell activation, which may promote tumor desmoplasia and progression [65].

Notably, poorly differentiated tumors exhibit an ascending gradient of stromal Gal-1 staining, indicative of its probable role in tumor aggressiveness or dedifferentiation [72]. Although initial research highlighted Gal-1’s predominance in the tumor stroma, emerging studies have pinpointed its presence in cultured human pancreatic cancer cells [70,73]. A particular focus has been on the cell membrane localization of Gal-1 during cell migration [73]. Insights from specific mouse models have further identified robust Gal-1 staining in both tumor epithelia and stromal compartments [74]. Such findings hint at a possible reciprocal interaction between pancreatic tumor cells and their stromal environment. Specifically, the redistribution of Gal-1 to the cell membrane in epithelial cells upon stromal interaction could facilitate cell–cell communication or promote invasive tendencies [67].

Furthermore, the conspicuous expression of Gal-1 in the stromal milieu, potentially originating from its secretion by pancreatic cancer cells, might suggest a paracrine modulation of the desmoplastic reaction characteristic of PDAC [73]. Overall, the intricate expression patterns and localization dynamics of Gal-1 in PDAC highlight its significance in the disease’s biology and the potential avenues for therapeutic interventions.

#### 4.1.1. Galectin-1 mRNA Expression in Pancreatic Cancer Tissue

Analyses of Gal-1 mRNA levels within pancreatic cancer tissues have unveiled stark contrasts when juxtaposed with healthy controls [68]. Employing Northern blotting techniques, pancreatic cancer specimens exhibited markedly amplified Gal-1 mRNA levels [68]. Quantitative assessments discerned that the mRNA concentrations in these malignant samples were approximately 11.9-fold that of non-cancerous tissue, affirming the strong linkage between elevated Gal-1 mRNA expression and the intricacies of pancreatic cancer progression [68].

#### 4.1.2. Galectin-1 Protein Expression in Pancreatic Cancer Tissue

Quantitative analysis reveals a substantial surge in Gal-1 protein concentrations in pancreatic cancer specimens, registering about 8.6-fold the levels seen in typical pancreatic tissues [68]. Detailed assessments identified pronounced Gal-1 immunoreactivity within the fibrous bands of tissue encapsulating the tumor, markedly surpassing the faint signals in regular pancreatic stroma [68]. It is intriguing that despite such profound stromal expression, pancreatic cancer cells largely remained devoid of Gal-1, echoing patterns observed in certain other cancers, including colorectal carcinomas [58].

#### 4.1.3. Serum Levels of Galectin-1 in Pancreatic Ductal Adenocarcinoma

Galectin-1 has been identified as a small soluble molecule that can be secreted into the extracellular space via a non-canonical secretory pathway [52]. In the context of cancer, including pancreatic ductal adenocarcinoma, increased levels of Gal-1 have been detected in both plasma and serum, in conjunction with overexpression in tumor tissues [22].

In assessing PDAC patients, plasma concentrations of Gal-1 are markedly higher than those observed in healthy controls, as quantified using Enzyme-Linked Immunosorbent Assay (ELISA) [22]. With a delineated cut-off value set at 17.7 ng/mL, there is a harmonized balance between diagnostic sensitivity and specificity, championing the utility of Gal-1 as a potential standalone biomarker for PDAC identification [22]. When juxtaposed against the benchmark tumor marker Carbohydrate Antigen 19-9 (CA19-9), Gal-1 showcases comparable diagnostic sensitivity and specificity. Leveraging both markers in tandem—Gal-1 and CA19-9—has been proposed to diminish the incidence of false negatives in PDAC diagnosis, thus bolstering the precision of preliminary CA19-9 assessments [22].

## 5. Galectin-1 Expression and Clinical-Pathological Parameters in Pancreatic Cancer

In the context of pancreatic cancer, Gal-1 overexpression has displayed a complex relationship with various clinical–pathological parameters. Although no significant correlations were found between Gal-1 overexpression and general patient characteristics, histopathological tumor parameters, tumor stage, or postoperative survival, a trend was identified [68]. Specifically, higher expression levels of Gal-1 were observed in more dedifferentiated cancer samples (G3), suggesting a potential relationship with tumor differentiation [68].

More notably, the expression of Gal-1 in the stromal region of PDAC tissues has been associated with multiple negative factors, serving as an indicator of unfavorable prognosis. These factors include larger tumor size, lymph node metastasis, perineural invasion, and poorer differentiation [70]. Elevated Gal-1 expression in the stromal compartment of PDAC has been further linked with perineural invasion and poor prognosis in patients, emphasizing the potential clinical relevance of Gal-1 as a biomarker in pancreatic cancer [72].

### 5.1. Galectin-1 and Lymph Node Metastasis in Pancreatic Cancer

Galectin-1 expression in pancreatic cancer demonstrates a significant association with the disease’s progression and metastasis, particularly in lymph node metastases [75]. Compared to normal pancreatic tissues, Gal-1 expression has been found to be substantially higher in both primary malignant tumors and metastatic tissues [75]. This increase appears even more pronounced in advanced lymph node metastases, highlighting a potentially critical role for Gal-1 in pancreatic cancer’s invasive nature [72,75].

The mechanisms underlying this association seem multifaceted. The high expression of Gal-1 in stromal cells, including activated PSCs, is known to induce epithelial–mesenchymal transition (EMT) in cancer cells [72]. This induction facilitates the acquisition of a metastatic phenotype, further characterized by elevated Gal-1 levels in cancer cells themselves. As a result, these cells exhibit enhanced invasive and metastatic abilities, promoting tumor progression [75].

### 5.2. Galectin-1 Expression and Prognosis in Pancreatic Cancer

Galectin-1 has gained prominence as a significant prognostic factor in pancreatic cancer, with various facets of its expression shedding light on patient outcomes [72]. One critical observation is the link between survival rates and Gal-1 expression levels [68]. Specifically, patients who exhibit robust Gal-1 expression tend to have a shorter overall survival span—a median duration of 14.1 months—compared to the 24.8 months observed in those showcasing weak expression [76]. Delving deeper into the statistical implications, a multivariate survival assessment applying the Cox proportional hazard model pinpointed pronounced Gal-1 expression as an independent harbinger of unfavorable prognosis, manifesting a relative risk of 4.676 [76].

Furthermore, insights from quantitative proteomic profiling have underscored Gal-1’s detrimental role concerning survival rates [76]. The intensity of Gal-1 staining, particularly in cancer-associated stromal cells, emerges as a pivotal parameter [72]. When considered alongside factors such as the tumor category and lymph node implications, it steadfastly holds its ground as an independent prognostic determinant [72].

Adding to this narrative, a comprehensive meta-analysis has echoed these findings [77]. It accentuates the strong association between elevated Gal-1 concentrations and a decline in overall survival for pancreatic cancer patients, suggesting a hazard ratio (HR) of 4.77 [77]. Another intriguing avenue of investigation revolves around Gal-1’s potential as a predictor of extended survival in pancreatic carcinoma cases, with preliminary results indicating a sensitivity rate of 64% and a specificity rate of 90% [78].

## 6. The Multifaceted Role of Galectin-1 in Pancreatic Ductal Adenocarcinoma (PDAC) Progression and Pathways

Galectin-1, a salient member of the galectin family, has been identified as a vital modulator of multiple signaling pathways contributing to the progression of pancreatic ductal adenocarcinoma [67,68,70,73,79,80,81,82]. This molecule’s complex involvement ranges from its significant effects on signaling pathways like ERK and Hedgehog-Gli to intricate interactions with molecules such as RAS, p16INK4a, MAPK, EGFR-Pdx1, and TGF-β1/Smad2 [67,70,80]. Recognizing the diverse roles of Galectin-1 could potentially unlock new therapeutic avenues to combat this aggressive cancer, especially given its impact on pancreatic cancer cell migration, fibrosis, and the epithelial–mesenchymal transition (EMT) [81]. For a concise overview of Galectin-1’s multifaceted involvement in PDAC progression and pathways, please refer to Table 3 provided below.

In the context of PDAC, Gal-1 plays a pivotal role in the proliferation of PSCs through its activation of the Extracellular Signal-Regulated Kinase (ERK) pathway [80]. Moreover, it enhances the production of chemokines such as Monocyte Chemoattractant Protein-1 (MCP-1) and Cytokine-Induced Neutrophil Chemoattractant-1 (CINC-1), primarily mediated by the Nuclear Factor-kappa B (NF-κB) signaling pathway, although the c-Jun N-terminal Kinase (JNK) and ERK pathways also contribute [80]. On the extracellular front, Gal-1 amplifies chemokine production by upregulating the mRNA expression of MCP-1 and CINC-1 in response to its concentrations [80]. Importantly, this effect can be countered by thiodigalactoside (TDG), a well-established inhibitor of Gal-1 [80]. The potential significance of this mechanism in the pathogenesis of PDAC lies in the critical roles chemokines play in tumor progression [80]. MCP-1 and CINC-1, in particular, are known to modulate the tumor microenvironment by recruiting immune cells, such as monocytes and neutrophils, which can further enhance inflammation and promote tumor growth and metastasis [83,84]. The heightened inflammation might also foster an environment conducive to angiogenesis and suppression of the anti-tumor immune response [85]. Additionally, the NF-κB, JNK, and ERK pathways are all crucial regulators of inflammation, cell growth, and survival [86,87]. Thus, the overexpression or dysregulation of these pathways, stimulated by Gal-1, could be instrumental in the aggressive behavior of PDAC.

The Hedgehog-Gli (Hh-Gli) pathway is a pivotal signaling mechanism, particularly in the realm of pancreatic carcinogenesis, playing a crucial role in cell proliferation, differentiation, and tissue patterning [88]. Against this backdrop, in-depth investigations involving PANC-1 cells have spotlighted Gal-1’s interaction with this critical pathway [67]. Gal-1 exerts regulatory control over genes essential for cell migration, adhesion, and malignant transformation, many of which intersect with the Hh-Gli pathway [67]. By modulating Gal-1 levels, researchers observed discernible effects on the Hh-Gli signaling components: diminished levels curtailed the pathway’s activity, while elevated Gal-1 levels amplified Gli transcriptional activities [67]. These insights emphasize the instrumental role of Gal-1 in modulating the Hh-Gli pathway and underscore its profound implications for PDAC progression.

The RAS pathway stands out in the progression of PDAC [89]. Within this framework, Gal-1 has demonstrated a substantial binding affinity to H-Ras and, to a lesser extent, K-Ras, suggesting its potential as a pivotal figure in the progression of pancreatic cancer by modulating the Ras-signaling pathway [90].

The tumor suppressor p16INK4a plays a pivotal role in the regulation of Gal-1 expression, unveiling a crucial interplay within the cellular milieu [91,92,93]. When p16INK4a is present, there is a marked increase in Gal-1 levels, coinciding with a heightened expression of the α5β51 integrin on the cell surface. This, in turn, instigates significant alterations in cell glycosylation patterns [91,92,93]. One of the most consequential outcomes of this interaction is its impact on anoikis, a specialized form of programmed cell death. Anoikis acts as a safeguard mechanism, preventing detached cells from colonizing elsewhere and forming harmful growths [94]. The modulation of Gal-1 levels by p16INK4a, along with the subsequent changes in cellular behaviors, plays a pivotal role in the progression of pancreatic ductal adenocarcinoma.

The EGFR-Pdx1 axis is increasingly recognized as a critical component in the molecular landscape of PDAC [67]. The Epidermal Growth Factor Receptor (EGFR) and Pdx1 transcription factor both have pivotal roles in pancreatic cell differentiation and proliferation [95,96]. Their dysregulation can drive the initiation and progression of PDAC, making the EGFR-Pdx1 axis a vital point of investigation for understanding the disease’s pathogenesis [97]. Within this intricate network, Gal-1 stands out as a consequential modulator. Observations have consistently shown that when Gal-1 expression is diminished, there’s a notable decrease in both EGFR and Pdx1 RNA levels in tumors [67]. This suggests a potential regulatory relationship, with Gal-1 serving as an upstream influencer on the EGFR-Pdx1 axis. By modulating the expression or activity of these key proteins, Gal-1 may directly or indirectly impact the trajectory of PDAC.

Lastly, Galectin-1’s interaction with the TGF-β1/Smad2 pathway in PSCs promotes a feedback loop likely expediting fibrosis [79]. Targeting Gal-1 could, thus, serve as a novel strategy for halting fibrosis in pancreatic cancer, emphasizing its critical influence on disease progression.

### 6.1. Galectin-1’s Influence on Pancreatic Cancer Cell Migration

Galectin-1’s role in enhancing the migratory potential of pancreatic cancer cells, particularly through the modulation of stromal cell-derived factor-1 (SDF-1), has been delineated [81]. Within PSCs, the predominant fibroblastic population implicated in the fibrotic responses of PDAC, Gal-1 directly governs the secretion of SDF-1 [81]. The downregulation of Gal-1 in these PSCs notably attenuates PDAC cell metastasis, underscoring the therapeutic promise of targeting Gal-1. Intriguingly, this regulatory mechanism pivots on the activation of NF-κB within PSCs, which subsequently boosts SDF-1 synthesis [81].

SDF-1, with its primary expression localized to PSCs, plays a consequential part in pancreatic cancer advancement [98]. When SDF-1 engages with its receptor, CXCR4, predominantly present on pancreatic cancer cells, it modulates tumor cell proliferation, malignancy, and resistance to chemotherapy. Notably, the interruption of the SDF-1/CXCR4 signaling axis via CXCR4 antagonists diminishes the tumor cells’ migratory and invasive prowess, underscoring the axis’s centrality in PDAC progression [98].

### 6.2. Galectin-1 and EMT in Pancreatic Carcinoma

Galectin-1 holds a significant role in the metastatic evolution and disease trajectory of PDAC [60]. When Gal-1 was targeted for suppression using small interfering RNA (siRNA), a pronounced decline in its expression within pancreatic cancer cells ensued. This resulted in a tempered migration and invasion potential of these cells [75]. A notable shift in cellular adhesion dynamics further accentuated Gal-1’s critical influence on the motility of cancer cells [75].

Epithelial–mesenchymal transition (EMT) represents a key cellular program where epithelial cells, typically characterized by strong cell-to-cell adhesion and apical–basal polarity, undergo transformation [99]. They shed these epithelial traits and embrace mesenchymal features, which include an enhanced migratory capacity, invasiveness, and resistance to apoptosis [100]. In the context of cancer, EMT is particularly alarming as it endows tumor cells with these aggressive traits, facilitating metastasis [101].

Upon Gal-1 inhibition, there was a clear reduction in markers characteristic of EMT. Key transcription factors such as Snai1 and Twist, which orchestrate the EMT process, were subdued [75,102]. Concurrently, the expression of MMP-10, an enzyme instrumental to tissue remodeling and cancer invasion, was mitigated [75]. This offers compelling evidence of Gal-1’s potential in initiating EMT, possibly in collaboration with MMP-10 and via the p38 MAPK pathway [75,102]. In an intriguing observation, the elevated expression of Gal-1, especially from PSCs, corresponded with increased levels of mesenchymal markers. Conversely, there was a reduction in epithelial markers [102]. This molecular rearrangement underscores the notion that PSC-derived Gal-1 actively promotes PDAC invasion by amplifying the EMT phenomenon.

When PANC-1 cells were co-cultured with PSCs that overexpressed Gal-1, these cells underwent marked phenotypic alterations, suggesting a mesenchymal transition [102]. The enhanced invasive attributes of PANC-1 cells in this setup were further reflected by molecular markers, reinforcing the idea that PSC-sourced Gal-1 augments tumor invasiveness. Interestingly, this influence of Gal-1 on the EMT process might have its roots in the NF-κB signaling pathway, given the observed fluctuations in associated molecular markers when PANC-1 cells interacted with PSCs harboring variable Gal-1 concentrations [102].

## 7. Galectin-1 in the Tumor Microenvironment of Pancreatic Cancer

In pancreatic cancer, Gal-1 originates predominantly from tumor fibroblasts and stellate cells [68]. While these cells secrete Gal-1 that binds to the extracellular matrix (ECM), pancreatic tumor cells also express and secrete Gal-1, contributing to its stromal presence [103]. The paracrine uptake of this protein by endothelial cells further amplifies its reach and effects [67]. In Figure 2, we visualize this intricate interplay where Gal-1 orchestrates a series of events in the PDAC microenvironment, culminating in the profound desmoplastic reaction and subsequent ischemic conditions, thereby emphasizing the multifaceted role of Gal-1 in pancreatic cancer progression and microenvironment modulation.

Gal-1 plays an instrumental role in the tumor microenvironment of PDAC [70]. Beyond stimulating chemokine production and accelerating PSC proliferation, it also amplifies the synthesis of key matrix components like collagen and fibronectin [80,104]. This drives a pronounced desmoplastic reaction, characterized by heightened stromal deposition and ECM remodeling, facilitating tumor growth and immune evasion [68]. When overexpressed in fibroblasts, Gal-1 not only accentuates the desmoplastic response but also fortifies the fibrotic barriers that hinder immune cell infiltration and compromise chemotherapy efficacy [103]. The depletion of Gal-1 in specific pancreatic tumors, in fact, results in reduced desmoplasia, underscoring its critical influence in shaping the tumor milieu [67].

### Gal-1’s Influence on Angiogenesis in PDAC

Pancreatic ductal adenocarcinoma displays a dense vasculature with poor perfusion and diminished integrity [105]. Despite its abundance, this vascular network fails to adequately deliver nutrients, oxygen, and drugs, largely due to the high interstitial fluid pressures stemming from fibrous interstitial fluid, which compresses blood vessels and hampers effective drug penetration [7,106].

Galectin-1 plays a crucial role in fostering angiogenesis in PDAC, particularly through its interaction with the extracellular matrix [67]. This angiogenic promotion is intricately tied to its influence on tumor cell metabolism. Central to this influence is the “Warburg effect”, wherein tumors preferentially undergo glycolysis even in oxygen-rich environments. Through modulation of this metabolic shift, Gal-1 not only enhances nutrient supply via angiogenesis but also indirectly supports tumor immune evasion [82,107,108].

The hypoxic tumor microenvironment, orchestrated by hypoxia-inducible factor-1 (HIF-1), stabilizes Gal-1 expression in conjunction with glucose transporter-1 (GLUT1) [82,109]. This dynamic fosters not only angiogenesis but also enhances tumor proliferation and metastatic potential. Within this context, Gal-1, similar to its counterpart galectin-3, propels glycolysis via pathways like the Phosphoinositide 3-kinase (PI3K) signaling cascade [82]. The resultant spike in lactate levels, emerging from glycolysis, acidifies the tumor environment [110]. This acidity not only promotes angiogenesis but also lures immunosuppressive components [110]. By enhancing the activity of enzymes linked to glycolysis, Gal-1 augments lactate production, facilitating the tumor’s adaptation to hypoxic conditions, which in turn promotes angiogenesis and suppresses immune responses [82].

Empirical evidence solidifies Gal-1’s significance in angiogenesis [90]. Enhanced Gal-1 expression in the tumor vasculature is associated with increased endothelial cell proliferation and migration, a relationship potentially modulated by H-Ras signaling [90]. Conversely, the diminished presence or absence of Gal-1 hampers these angiogenic processes [67]. Studies on Gal-1 deficient mice further support this, showcasing stunted tumor growth attributed to impaired angiogenesis [111]. In specific mouse models, lower Gal-1 levels were associated with reduced angiogenesis and decreased intraperitoneal hemorrhages, further underscoring Gal-1’s essential role in angiogenesis and the broader spectrum of tumor progression [67].

## 8. Galectin-1 and Pancreatic Stellate Cells in PDAC Fibrosis

Pancreatic Stellate Cells are pivotal in engendering stromal fibrosis in PDAC [25]. They achieve this by remodeling the ECM and releasing a range of compounds, such as IL-1, IL-6, IL-8, IL-10, VEGF, Platelet-Derived Growth Factor (PDGF), and TGF-β, which play integral roles in collagen synthesis, and fostering ECM deposition [79,112].

Galectin-1 is a characteristic product released by PSCs and plays a significant role in enhancing the function of these cells [113]. Laboratory investigations have shown that Gal-1 can boost the growth of PSCs and their production of collagen [80]. Interestingly, substances produced by PSCs also appear to stimulate the release of more Gal-1 [19]. This suggests a mutual reinforcement between Gal-1 and PSCs, where each enhances the activity of the other.

Within the context of PDAC, Gal-1 plays a crucial role in directing PSCs to produce cytokines like IL-6, IL-10, and IL-8 [82]. This action in turn speeds up the deposition of ECM, a process vital for tumor development. In animal studies, Gal-1 has been observed to enhance PSC-driven IL-10 production, resulting in a more fibrous tumor environment [113]. Additionally, under low-oxygen (hypoxic) conditions, Gal-1 further magnifies its influence by amplifying HIF expression, prompting PSCs to have an even greater fibrotic impact [114]. Studies in mice that lack Gal-1 have shown disruptions in the tumor stroma related to the Hedgehog (Hh) signaling pathway, emphasizing Gal-1’s critical contribution to the fibrosis seen in PDAC [67].

Gal-1 is integral to the inflammatory responses that contribute to fibrosis in pancreatic cancer [115]. Notably, it catalyzes the activation of the NF-κB pathway, thereby intensifying inflammation. Such heightened inflammation propels fibrosis, which is primarily orchestrated by PSCs [80]. As these PSCs navigate the tumor microenvironment, they engage with immune cells, including macrophages and B lymphocytes. This cellular crosstalk magnifies the fibrotic landscape characteristic of PDAC [116]. In addition, Gal-1 disrupts the immune equilibrium between Th1 and Th2 cells in pancreatic malignancies. An outcome of this disruption could be a surge in IL-5, a cytokine implicated in the progression of tumor-associated fibrosis [113,117].

Studies have highlighted a notable increase in Gal-1 expression in PSCs from PDAC samples when compared to normal pancreatic tissues [68]. When Gal-1’s activity is deliberately curbed in human PSCs, these cells display diminished migration and invasive tendencies [118]. In contrast, introducing Gal-1 externally encourages cell growth and boosts collagen production in rat PSCs. These effects hint at the involvement of specific cellular signaling mechanisms, including the MEK1/2-ERK1/2 pathways, as well as the engagement of transcription factors like NF-κB [80,104]. With the accumulating insights, it becomes apparent that Gal-1, especially when expressed by activated PSCs, could serve as a potent target for immunotherapeutic strategies in treating pancreatic cancer.

### 8.1. Autocrine Effect of Gal1 on PSC Activation in PDA

Pancreatic stellate cells, when activated, play a significant role in the progression of pancreatic ductal adenocarcinoma [25]. As these PSCs become active, they undergo noticeable changes: they lose their fat droplets and show increased levels of α-smooth muscle actin (α-SMA), which is vital for the process of fibrogenesis [119]. Alongside these changes, there’s a marked rise in Gal-1 expression, which transforms the appearance of these cells, making them resemble fibroblasts [70].

Gal-1, found in activated PSCs, plays a vital role in maintaining their active and fibrogenic characteristics [70]. Interestingly, when the levels of Gal-1 are reduced in human PSCs, these cells tend to become less active, almost returning to a dormant state [104]. This suggests that Gal-1 is crucial for keeping PSCs activated and supporting their transformation into fibroblast-like cells.

Gal-1 exerts a significant autocrine effect on PSCs, shaping their behavior and pivotal role in the progression of pancreatic cancer. Specifically, when Gal-1 binds to its ligands, it sets off a cascade of cellular signals [74]. A notable pathway influenced by this binding is the ERK pathway, crucial for fostering cell growth and ensuring survival [19]. Another is the TGF-β1/Smad signaling, which, when activated by Gal-1, propels PSCs into an enhanced state of activity [79]. In tandem, these signaling pathways dictate the transformation of PSCs, positioning them as central players in pancreatic cancer progression [19]. Delving into how Gal-1 modulates these pathways and the activation of PSCs provides a deeper understanding of the intricate interplay between PSCs and Gal-1 within the context of PDAC. Acquiring such insights is fundamental in demystifying the root causes and charting prospective treatment strategies for pancreatic cancer.

### 8.2. Paracrine Secretion of Gal-1 by PSCs in Pancreatic Cancer

Gal-1, secreted by activated PSCs, exerts powerful paracrine effects on surrounding cells, especially on surrounding pancreatic tumor cells and the immune milieu [19]. To put it simply, a paracrine effect means that a molecule produced by one cell can influence the behavior of nearby cells. Evidence from laboratory investigations has underscored that exposure to recombinant Gal-1 (rGal-1) accentuates the invasiveness of specific pancreatic cancer cell lines, such as CFPAC-1 [120]. Concurrently, media conditioned by human PSCs have been shown to bolster the migratory and invasive tendencies of the RWP1 pancreatic cancer cell lines—a phenomenon that attenuates significantly upon depletion of Gal-1 [74].

On a molecular echelon, the paracrine actions orchestrated by Gal-1 induce a cascade of cellular modifications within pancreatic carcinoma cells [74]. Notably, there’s an upregulation of proteins like vimentin, coupled with a concomitant downregulation of E-cadherin, fostering an environment conducive for increased cellular migration and invasion [102]. Moreover, Gal-1, stemming from PSCs, amplifies the proliferative capacity of pancreatic tumor cells [72]. This is exemplified by enhanced cell multiplication when these tumor cells are co-cultured with PSCs exuberantly expressing Gal-1, in stark contrast to their counterparts with diminished Gal-1 levels [72].

### 8.3. Gal-1’s Influence on PSC-Mediated MMP Activation in PDAC

In the intricate milieu of pancreatic malignancy, the role of Gal-1, predominantly secreted by PSCs, emerges as vital, especially in the orchestration of matrix dynamics [113]. Elevated Gal-1 expression in PSCs augments the secretion of matrix metalloproteinases (MMP-2 and MMP-9) [120]. These enzymes, known for their proteolytic functions, are instrumental in degrading the extracellular matrix (ECM) and cellular basement membrane (BM), thus facilitating tumor progression and invasion capabilities [121]. A notable observation was that of CFPAC-1 cells, which, when in co-culture with PSCs, showcased an increased MMP-2 and MMP-9 synthesis, coupled with accentuated proliferation and invasiveness [120]. This effect was pronounced when interacting with cancer-associated PSCs (CaPSCs) as opposed to their normal counterparts (NPSCs) [72]. However, this amplified activity seemed to diminish with the introduction of lactose, a recognized inhibitor of galectins [72,120].

Delving further into the molecular intricacies, an engaging interplay between MMP-2, Tissue Inhibitor of Metalloproteinase-1 (TIMP-1), and Transforming Growth Factor-beta 1 (TGF-β1) was discerned [79]. Gal-1’s interaction with PSCs led to a discernible increase in MMP-2 levels and an even more substantial rise in TIMP-1 levels. This modulation suggested Gal-1’s potential role in encouraging ECM protein synthesis, an indication of its pro-fibrogenic tendencies [79]. To understand in greater detail, MMP-2 aids both in the proliferative activity and the degradation of type IV collagen present in the basement membrane [122]. Yet, the overarching expression of TIMP-1 serves as a counteracting force. This dynamic pushes the balance between ECM synthesis and degradation towards fibrogenesis, illuminating Gal-1’s integral role in the genesis of pancreatic fibrosis [123].

### 8.4. Gal-1’s Interplay with CAF in PDAC Development

In the intricate landscape of PDAC, cancer-associated fibroblasts (CAFs) play a quintessential role in fostering tumor fibrosis [124]. These CAFs, constituting a significant portion of the ECM, are primarily driven to activation under the influence of cytokines and agents such as TGF-β, IL-6, and IL-10 [125,126]. Upon activation, their physiological response leads to the secretion of matrix metalloproteinases (MMPs) [127]. This secretion acts as a catalyst for the degradation of the basement membrane, ensuing in comprehensive ECM remodeling. Concurrently, these fibroblasts potentiate fibrosis via the release of hyaluronic acid and collagen [128].

Delving into the relationship between Gal-1 and CAFs, it is evident that Gal-1 emerges as a significant pro-fibrotic mediator, meticulously orchestrating CAF activation. Particularly noteworthy is its ability to enhance IL-6 and IL-10 secretion in the PDAC milieu, thereby accelerating the activation of CAFs which in turn amplifies stromal tumor fibrosis. Insights from murine studies also spotlight Gal-1’s potential in precipitating CAF activation via the Hedgehog signaling pathway [67]. Moreover, within the domain of PSCs, a surge in Gal-1 expression has been linked to an indirect activation of CAFs—this is predominantly mediated by an upregulated TGF-β expression, leading to a subsequent rise in MMP secretion and profound changes in the ECM [82].

## 9. Gal-1 and Immune Evasion in Pancreatic Ductal Adenocarcinoma (PDAC)

In the intricate realm of immunology, Gal-1 emerges as a molecule of paramount significance, endowed with robust immunoregulatory attributes [54]. It has been observed to exert a profound influence on neutrophil dynamics by curtailing their migration, thereby altering the innate immune response [103]. Furthermore, its role in macrophage plasticity is noteworthy; specifically, Gal-1 facilitates the transition of macrophages from the M1 phenotype, which is inherently pro-inflammatory, to the M2 phenotype, characterized by its immunosuppressive properties. This transformation, when contextualized, points toward Gal-1’s overarching theme of tempering immune responses, as seen by its propensity to encourage the maturation of tolerogenic dendritic cells and to foster T-regulatory cell differentiation [82].

Delving into the context of pancreatic cancer, the role of Gal-1 becomes even more salient. Within the tumor microenvironment, it acts as a pivotal architect in establishing an immunosuppressive milieu [113]. One of the salient manifestations of this is seen in how Gal-1 deters T cell activation, tipping the scales towards an immunological state that is less reactive [74]. Gal-1 plays a noteworthy role in suppressing the immune response against tumors. Specifically, it acts by inducing apoptosis in T cells, which are vital components of the immune system responsible for recognizing and attacking tumor cells. By promoting the death of these T cells, Gal-1 effectively weakens the immune system’s ability to identify and combat tumor cells [129]. Beyond this direct impact on immune cells, Gal-1 also exerts influence on the broader immune environment. It actively modulates the secretion patterns of cytokines, which are signaling molecules that dictate immune responses. By altering these secretion patterns, Gal-1 creates an anti-inflammatory environment, favoring tumor survival and growth, and this modulation affects a wide range of immune cells, further emphasizing its significant role in cancer progression [113].

In the broader perspective of pancreatic pathologies, early indications of Gal-1 expression have been associated with stromal activation events such as inflammation and preneoplasia [22,130]. Its tangible presence in the stroma during ailments like pancreatitis or preceding malignant transformations signifies its role in creating an immune sanctuary [22]. This niche environment is conducive for immune evasion, laying a fertile ground for tumor inception and progression. Drawing a nexus between Gal-1-mediated stroma activation and the attainment of immune privilege paints a comprehensive picture of its centrality in pancreatic maladies, highlighting the molecule’s potential as a therapeutic target. Following this intricate interplay, Figure 3 provides a visual representation of the stages and processes where Gal-1 takes center stage, emphasizing its pivotal role in stromal activation, immune evasion, and the subsequent evolution of pancreatic cancer.

### 9.1. Galectin-1 Influence on T Cell Dynamics in PDAC

Acquired immunity plays a pivotal role in pancreatic pathologies, orchestrating complex interactions that influence disease progression and outcomes [131]. Galectin-1, a β-galactoside-binding protein, profoundly modulates T cell function in both physiological and pathological states [132]. Its influence is particularly pronounced in PDAC, where it emerges as a facilitator of tumor immune evasion [82]. Within the PDAC environment, Gal-1 takes center stage as a key agent promoting tumor immune evasion [82]. One of its striking features is its ability to curb T cell infiltration [103]. Specifically, while CD3+ T cells can be observed in the vicinity of malignant cells, they face a formidable barrier in penetrating the tumor parenchyma [108]. This suggests a significant role for Gal-1-expressing PSCs in shaping the tumor-immune landscape, ostensibly by impeding T cell access to the tumor core [103].

Delving deeper into the molecular intricacies of Gal-1, its adeptness in orchestrating T cell apoptosis is evident [76]. The molecule demonstrates multifaceted mechanisms in this regard. Its interaction with the CD45 molecule on T cells culminates in the degradation of fodrin, setting the stage for T cell phagocytosis and consequent apoptosis [133,134]. Moreover, the binding of Gal-1 to receptors such as CD7 and Complement receptor 3 (CR3) augments this apoptotic cascade [135,136]. Notably, in the presence of Gal-1, quiescent T cells are propelled towards cell death via pathways such as the fas/caspase-8 axis, and intriguingly, Gal-1 can even induce the release of molecules like ceramide, which can activate other apoptosis circuits, including those involving caspase-9 and caspase-3 [137,138]. There is also an underpinning of the JNK/C-Jun/AP-1 pathway in Gal-1-mediated apoptosis [139].

Lastly, Gal-1’s influence is not confined to direct cellular interactions but also extends to the modulation of the cytokine milieu in PDAC. The overexpression of Gal-1, particularly in PSCs, effects a paradigm shift in cytokine dynamics: it diminishes the expression of Th1 cytokines, such as IL-2 and INF-γ, while concurrently promoting Th2 cytokines like IL-4 and IL-5 [113]. This cytokine modulation, pivotal in shaping immune responses, can be partially reversed by attenuating Gal-1 levels [113]. A case in point is the observed suppression exerted by endogenous Gal-1 from PSCs on Th1 cytokines, a phenomenon particularly pronounced in human cancer-associated PSCs (hCaPSCs) [76]. Importantly, this shift from a Th1 to Th2 cytokine profile holds profound implications for tumor immunity, fostering an environment more amenable to tumor progression and potentially undermining anti-tumor immune responses.

### 9.2. Galectin-1 and NK Cell Dysregulation in PDAC

Natural Killer (NK) cells are innate immune effectors with the inherent ability to eliminate malignant, aged, and virus-infected cells through various cytokines and proteins [140]. However, within the hostile environment of PDAC, these formidable defenders become vulnerable due to the overexpression of Gal-1 [82].

Within the complex milieu of PDAC, Gal-1 plays an influential role, particularly concerning the modulation of NK cell activity [82]. Customarily, NK cell activation, which is facilitated by IL-2 and marked by the release of key entities such as Interferon-γ (IFN-γ) and tumor necrosis factor-α (TNF-α), is seen to be compromised in the presence of PDAC [141]. This anomaly is likely due to the ability of Gal-1 to curtail the production of IL-2, thus tipping the Th1/Th2 cytokine balance and diminishing NK cell activation [113]. Furthermore, Gal-1 seems to enhance IL-6 secretion, a factor which is known to negatively impact NK cell functionality [142].

Obesity, an often-discussed risk factor for PDAC, has been linked to elevated levels of Gal-1, resulting in increased adipocyte infiltration. This raised expression of Gal-1 coincides with heightened IL-6 levels, which in turn leads to a reduction in IFN-γ production, thus hindering NK cell operations [142]. Additionally, the presence of elevated Gal-1 also seems to promote the secretion of matrix metalloproteinase (MMP9) and indoleamine 2,3-dioxygenase (IDO), both of which are recognized for their inhibitory effects on NK cell efficiency [143]. The correlation between increased Gal-1 secretion and type-2 diabetes, commonly observed in PDAC patients, presents a significant concern [144]. This relationship intensifies insulin resistance, typical of type-2 diabetes, and boosts Gal-1 production, further dampening NK cell functionality in the face of PDAC [145].

Lastly, attention must be directed towards NK Group 2, Member D (NKG2D), a critical NK cell surface receptor. This receptor, vital for NK cell activation, interacts with MHC class I-related molecules [146]. Yet, in PDAC conditions, its expression appears reduced, a phenomenon potentially influenced by the hypoxic conditions amplified by Gal-1. In this scenario, Gal-1 is believed to foster a hypoxic environment, thereby augmenting Hypoxia-Inducible Factor (HIF) activity [147]. This escalation seems to inhibit NKG2D expression, consequently weakening the combative capabilities of NK cells against tumor formations.

### 9.3. Galectin-1’s Role in TAM-Mediated Immune Evasion in PDAC

Tumor-associated macrophages (TAMs), specifically of the M2 phenotype, play a pivotal role in immune suppression within PDAC landscapes [36]. In synergy with myeloid-derived suppressor cells (MDSC), they inhibit T cell functionality and augment T cell apoptosis [148,149].

Notably, a defining characteristic of PDAC, the activation of hypoxia-inducible factor-1 (HIF-1), shows a pronounced association with TAM expression [150]. This association is further marked by an enhanced lactate production, underscoring the intertwined nature of these phenomena [82]. During hypoxic events, the HIF-1 pathway is observed to escalate Gal-1 concentrations, an effect that seems to be further amplified by the concurrent H-ras pathway [151]. This surge in Gal-1 notably stimulates the production of vascular endothelial growth factor (VEGF) through the upregulated secretion of interleukin-6 (IL-6) [152]. In another significant sequence, upon lipopolysaccharide (LPS) exposure, Gal-1 is known to instigate ADAM10/17 production, which subsequently gives rise to lactic acid [147]. This chain of events activates the nuclear factor erythroid 2-related factor 2 (Nrf2), influencing reactive oxygen species (ROS) generation and consequently steering TAM differentiation and augmenting VEGF secretion [152].

Moreover, an interesting interplay is evident when Gal-1 binds to Neuropilin1 (NRP-1) found on CAFs [153]. This association appears to facilitate VEGFR2 signaling, thereby dampening angiogenesis while enhancing the hypoxic attributes inherent to PDAC [154]. Such conditions are conducive for the activation of both TAM and MDSC, further strengthening the TAM-driven immunosuppression prevalent in PDAC. To encapsulate, Gal-1 appears to be a central figure in molding the immunosuppressive facet of PDAC, primarily through its influence on TAM activation and proliferation.

### 9.4. Gal-1’s Influence on MDSCs in Pancreatic Ductal Adenocarcinoma

In the realm of Galectin-1 functionalities, its profound influence over immune cell infiltration in PDAC cannot be understated [54]. A cellular element under its considerable purview is the MDSCs, entities renowned for their pronounced immunosuppressive characteristics [155]. Within the context of Kras-driven PDAC, tumor cells exhibit a propensity to release granulocyte-macrophage colony-stimulating factor (GM-CSF), a molecule instrumental in choreographing the MDSC recruitment [149,156]. The pivotal orchestration of Gal-1 in MDSC accumulation is further illuminated by experimental observations wherein Gal-1-devoid mice depict a marked reduction in MDSC presence [74]. It behooves us to comprehend the significance of MDSCs in PDAC, as their heightened presence often correlates with advanced tumor stages, underscoring the vital role they play in tumor progression and immune evasion [37].

### 9.5. Tolerogenic Dendritic Cell Modulation by Galectin-1 in PDAC

Galectin-1 emerges as a critical molecular entity in the landscape of PDAC, particularly in its influence on the differentiation of dendritic cells towards a tolerogenic disposition within the tumor microenvironment [157]. This pivotal differentiation underlines Gal-1’s cardinal role in fostering an immunosuppressive milieu within PDAC, thereby bolstering tumor-driven immune tolerance [74]. Tolerogenic dendritic cells, characterized by their unique attributes, not only play sentinel roles in PDAC but also act as mediators, bridging the chasm between reactive and suppressive immune responses intrinsic to the tumor ecosystem [158]. The potential diminishment or absence of Gal-1 might perturb these finely tuned mechanisms, resulting in a conceivable augmentation in effector T cell populations within the tumor setting [74]. This potential shift underscores the profound implications of Gal-1’s modulation on the immunological equilibrium of PDAC, highlighting its potential as a therapeutic target.

### 9.6. Gal-1’s Modulation of Neutrophils in Pancreatic Tumors

Neutrophils, occupying a cardinal position within the tumor microenvironment of PDAC, are subjected to notable regulatory modulation by Gal-1 [159]. This molecule exhibits a salient propensity to impede the chemotaxis and consequent migratory abilities of these neutrophils [160]. Conspicuously, Myeloperoxidase (MPO) staining elucidates an enhanced neutrophilic presence in ductal malignancies concomitant with Gal-1 abrogation [67]. Such empirical observations underscore the imperative function of Gal-1 in safeguarding the immune privilege inherent to pancreatic neoplasms, attenuating neutrophil-centric immunological activities, thereby accentuating its consequential role in the overarching therapeutic landscape of PDAC.

## 10. Gal-1 Inhibition in PDAC: Therapeutic Horizons

Gal-1, with its evident role in promoting immune evasion in pancreatic cancer, is being studied as a therapeutic target [82]. Depletion of Gal-1 has been linked with an increased influx of T cells and neutrophils in pancreatic tumors, shedding light on its potential to reinstate the body’s immunological defenses against cancer [103]. Existing data indicate that anti-Gal-1 strategies, like inhibitors or monoclonal antibodies, exhibit anti-proliferative and anti-angiogenic effects across multiple tumor types, underscoring their potential in countering Gal-1-induced immunosuppression in pancreatic tumors [66,161].

Notably, the observed viability and fertility in Gal-1-knockout mice suggest a possible safety of treatments targeting Gal-1 [103]. This makes Gal-1 an interesting possible target for immune treatments against pancreatic cancer. But moving from lab studies to real-life treatments is not simple. Galectins, including Gal-1, are complex and need careful study before making specific drugs. Aspects like protein conformation, receptor glycan repertoire, cell type, and cellular conditions can alter the inhibitory effects of Gal-1. Therefore, using results from animal tests directly in humans may not always work. This is because humans and animals can have differences in how these proteins are expressed, their specific roles, and how the immune system works. Thus, it is important to be careful and study well before making new treatments [103].

However, the substantial expression of Gal-1 in human pancreatic cancer and its demonstrable influence on immune response enhancement, tumor cell proliferation reduction, and tumor stroma modification solidify its position as a prospective therapeutic candidate. As we delve deeper into understanding Gal-1’s role in PDAC, it becomes increasingly evident that its inhibition offers promising therapeutic potential. To harness this potential fully, further research is imperative to refine the nuances of Gal-1 inhibition, enhance inhibitor specificity, and assess their effectiveness and safety in human trials. In this context, two avenues are emerging as particularly significant: the development of specific inhibitors and the promising intersection of Gal-1 with nano-oncology.

Inhibitors: lab trials have spotlighted a Galectin-1 inhibitor, LLS2, that augmented the efficacy of chemotherapy agent paclitaxel across multiple human cancer cell lines, encompassing pancreatic cancer cells [162].

Nanotechnology: Nano-oncology is carving out novel avenues to deploy galectin-based inhibitors or galectin ligand-augmented nanoparticles for diverse oncological applications. A recent study pivoted on Gal-1’s upregulation in pancreatic cancer for the delivery of magnetic nanoparticles to cancer tissues [162]. These nanoparticles were adapted with glycosylated peptides from tissue plasminogen activator (tPA), a protein overexpressed in PDAC and recognized as a Gal-1 ligand [162]. Mouse PANC-1 xenograft magnetic resonance imaging elucidated a superior uptake of these tailored nanoparticles relative to non-specialized ones [162]. This endorses Gal-1’s potential as a therapeutic vector in pancreatic cancer. Gal-1 directed nanoparticles might usher in groundbreaking strides in pancreatic cancer theranostics, merging diagnostic imaging with therapeutic intervention.

## 11. Gal-1 in PDAC: Current Challenges and Future Perspectives

Pancreatic ductal adenocarcinoma (PDAC) presents a labyrinth of molecular intricacies that has persistently challenged the scholarly efforts of oncological researchers. At the heart of these enigmatic mechanisms lies Galectin-1 (Gal-1), a protein increasingly acknowledged for its multi-dimensional influence on tumor genesis, evolution, and immune circumvention. As endeavors intensify to fathom its significance within PDAC, it becomes paramount to cognize the impending challenges and contemplate potential revelations that forthcoming investigations might herald.

### 11.1. Current Challenges

The exploration into the significance of Gal-1 in PDAC has been a complex endeavor, marked by numerous scientific and therapeutic obstacles. The undeniable presence of this protein in pancreatic tumors has been established, but fully grasping its specific role and turning this understanding into effective treatments poses substantial challenges.

Firstly, Gal-1’s multifunctionality in pancreatic cancer adds to its complexity as a target. Its roles range from fostering immune evasion and advancing tumor angiogenesis to altering tumor metabolism. Thus, it is essential to comprehend the exact mechanisms and the varied cellular contexts in which Gal-1 functions.

Another point of contention is the precision required in targeting Gal-1. Crafting therapeutic agents or inhibitors that can zero in on Gal-1 without disturbing other vital cellular pathways is a delicate task. Furthermore, while animal models have been indispensable in providing insights, there is always a degree of apprehension when applying these findings to humans. This is due to the inherent differences in lectin expression, immune responses, and cellular interactions between species.

Additionally, safety remains a paramount concern. Although Gal-1 knockout models have demonstrated both viability and fertility, the long-term effects of inhibiting Gal-1 in humans have not been extensively explored. It is vital to ensure that prolonged targeting of Gal-1 does not produce adverse outcomes. Lastly, the diverse nature of pancreatic tumors introduces another layer of complexity. With varying levels of Gal-1 expression and distinct interactive pathways, it becomes challenging to devise a one-size-fits-all therapeutic strategy.

### 11.2. Future Perspectives

As we navigate the emerging developments in PDAC research, we are hopeful about the potential innovations and strategies on the horizon. Galectin-1, or Gal-1, stands out as a cornerstone in these advancements, suggesting that our approach to pancreatic cancer management is on the cusp of significant transformation.

Tailored therapies: The era of precision medicine brings forth the tantalizing possibility of therapies tailored to individual needs. Specifically, there lies an opportunity to design treatments targeting Gal-1, which are finely tuned based on each patient’s unique tumor profile. Such specificity holds the promise to significantly boost therapeutic outcomes, ensuring more patients benefit from interventions.

Combination therapies: The idea of combining therapies is gaining traction. By inhibiting Gal-1 in tandem with other established treatment methods, such as chemotherapy or immunotherapy, we could amplify the overall anti-tumor effects, thereby enhancing the chances of treatment success.

Nano-oncological approaches: The fusion of nanotechnology with oncology presents an exciting avenue. Utilizing nanoparticles that specifically target Gal-1 might lead to a paradigm shift in how we deliver anti-cancer agents. This approach could amplify treatment efficacy while minimizing undesirable off-target effects.

Theranostics—A convergence of diagnostic and therapeutic avenues: The emerging field of theranostics offers a unique blend of diagnosis and therapy. Gal-1-targeted agents, in this context, hold potential to serve dual roles. They could act both as tools for diagnosing the disease and as therapeutic agents. This dual-function approach could significantly enhance early detection rates and optimize treatment response, providing a more integrated way to manage pancreatic cancer.

Biomarker potential: Gal-1’s conspicuous expression in pancreatic tumors suggests its potential role as a biomarker. It could prove invaluable for early disease diagnosis, prognosis assessment, and for monitoring how patients respond to therapies.

In essence, the synergy of molecular biology, nanotechnology, and precision medicine, all converging around the pivotal role of Gal-1, lights the way forward. For patients, clinicians, and researchers alike, it is a beacon of hope. While the journey ahead is undoubtedly complex, each step taken pushes the boundaries, offering a brighter future in diagnosing, treating, and understanding the prognosis of PDAC.

## 12. Conclusions

Pancreatic ductal adenocarcinoma (PDAC) represents a formidable challenge in oncology due to its aggressive nature, rapid metastasis, and resistance to conventional treatments. Central to this challenge is the protein galectin-1 (Gal-1), which has gained prominence not only as a potential diagnostic and prognostic tool but also as a therapeutic target. Gal-1’s profound influence on the tumor microenvironment, from modulating stromal interactions and fostering immunosuppression to driving tumor growth and progression, underscores its critical role in PDAC pathology. Its multifaceted interactions, whether by shaping the immune backdrop, influencing matrix dynamics, or directing tumor cell behavior, highlight its integral nature in this malignancy. Thus, while the potential of Gal-1 as a therapeutic target is evident, the path to its clinical exploitation is intricate. Embracing advances in molecular biology, nanotechnology, and precision medicine with a central focus on Gal-1 will be paramount in charting new therapeutic avenues, ultimately aiming to revolutionize pancreatic cancer management.

## Figures and Tables

**Figure 1 ijms-24-15500-f001:**
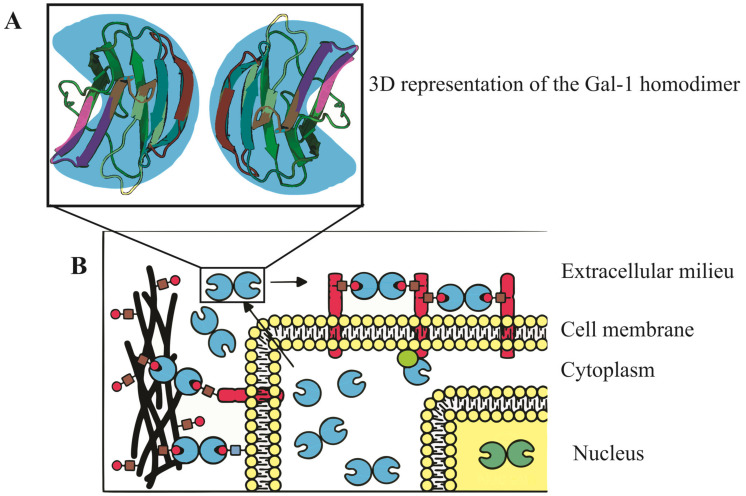
Three-dimensional representation and subcellular localization of Gal-1 homodimer. (**A**) A detailed 3D model of the Gal-1 homodimer, accentuating its β-sandwich configuration. This structure comprises two antiparallel β-sheets, the first containing five strands and the second with six. The intricate architecture of Gal-1 is key to its binding properties and biological functions. (**B**) Schematic illustration of Gal-1’s diverse cellular localizations, with depictions in the nucleus, cytoplasm, cell surface, and extracellular matrix. These varied locations emphasize Gal-1’s multifaceted roles in regulating vital cellular processes including cell cycle control, migration, adhesion, immune responses, apoptosis, and inflammation.

**Figure 2 ijms-24-15500-f002:**
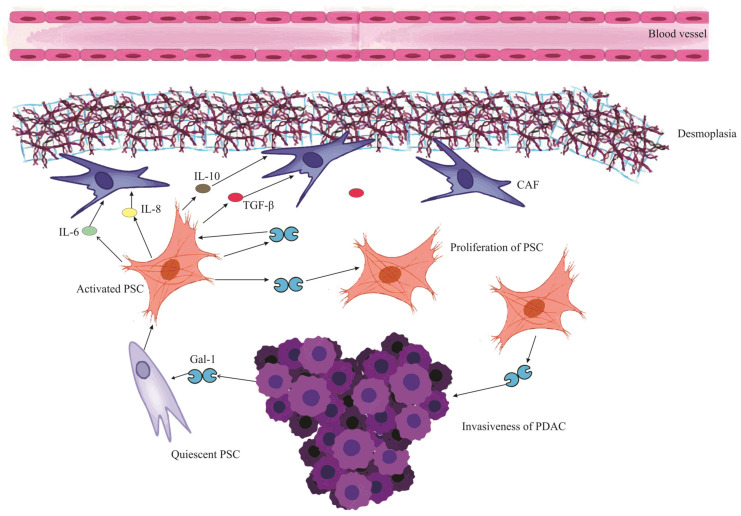
Mutual reinforcement between Galectin-1 and Pancreatic Stellate Cells in PDAC microenvironment. The schematic illustrates the cascade initiated when PDAC (Pancreatic Ductal Adenocarcinoma) releases Galectin-1 (Gal-1), which activates and stimulates the proliferation of PSCs (Pancreatic Stellate Cells). These activated PSCs subsequently release a myriad of mediators, including IL-6 (Interleukin-6), IL-8 (Interleukin-8), IL-10 (Interleukin-10), and TGF-β (Transforming Growth Factor-beta). These mediators contribute to a series of downstream effects: Remodeling of the ECM (Extracellular Matrix), Collagen Synthesis, Conversion to CAF (Cancer-Associated Fibroblast) cells, and fostering ECM deposition, cumulatively leading to a pronounced desmoplastic reaction. This desmoplasia contributes to the ischemic environment characteristic of PDAC. Furthermore, the activated PSCs themselves release Gal-1, signifying a mutual reinforcement loop between Gal-1 and PSCs. The impact of Gal-1 is also discerned in PDAC cells, where it heightens the invasiveness, fortifying their migratory and invasive tendencies.

**Figure 3 ijms-24-15500-f003:**
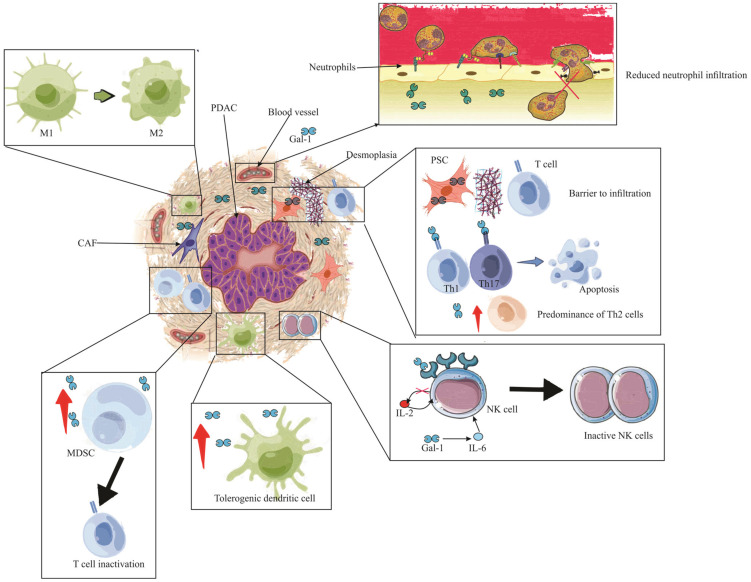
The immunosuppressive role of Gal-1 in PDAC. The illustration depicts the multifaceted impact of Gal-1 in mediating immune suppression and evasion within the PDAC microenvironment. Enhanced Gal-1 expression is shown to inhibit the chemotaxis and infiltration of neutrophils into PDAC tissue. The altered immune landscape is marked by a diminished T lymphocyte count, with a pronounced Th2 cytokine profile. Crucially, Pancreatic Stellate Cells (PSCs) expressing Gal-1 act as formidable barriers, obstructing T cell infiltration. Gal-1 further instigates apoptosis in Th1 and Th17 cells, reinforcing its immunosuppressive role. The detrimental effects extend to the Natural Killer (NK) cells, where Gal-1 is seen to inhibit and inactivate them, largely due to the compromised IL-2 production and elevated IL-6 expression. The scene reveals an augmented presence of tolerogenic dendritic cells and MDSCs, underscoring the immune-evasive strategies employed. Finally, the PDAC is encapsulated by an abundance of tumor-associated macrophages (TAMs) of the M2 phenotype, with neighboring MDSCs, highlighting their synergistic role in thwarting T cell functionality and bolstering apoptosis. Note: PDAC (Pancreatic ductal adenocarcinoma), PSC (Pancreatic Stellate Cells), IL (Interleukin), Gal-1 (Galectin-1), CAF (Cancer-associated fibroblasts), MDSC (Myeloid-derived suppressor cells), M1 (Type 1 macrophage), M2 (Type 2 macrophage).

**Table 1 ijms-24-15500-t001:** Gal-1’s impact on PDAC progression and pathology.

Factor	Influence of Gal-1
Stromal Interactions	Role in stromal presence, angiogenesis, and immune evasion
Tumor Growth	Impact on cell migration, EMT, and stromal cell-derived factor-1 (SDF-1) modulation.
Immune Landscape	Effect on neutrophils, macrophages, T cells, NK cells, and dendritic cells.
Tumor Fibrosis	Involvement in PSC activation and ECM remodeling.

**Table 2 ijms-24-15500-t002:** Key features of PDAC’s stromal and immune landscape.

Components	Description/Role	Clinical Implications
Stromal Desmoplasia	Extensive extracellular matrix; obstructs drug delivery, promotes hypoxia	Resistance to therapies; potential target to disrupt tumor-promoting environment
Pancreatic Stellate Cells	Interact with cancer cells; secrete cytokines, growth factors	Fuel tumor progression; paracrine signaling as therapeutic target
Immunosuppressive Components	M2 macrophages, myeloid-derived suppressor cells, Tregs	Impede T cell activation; limit CTL function
Secreted Immunosuppressants	TGF-β, IL-10, IL-6, VEGF, Fas ligand	Dampen immune response; reduce immunotherapy effectiveness
Tumor-infiltrating CTLs	Present but limited due to tissue resistance	Potential for targeted immunotherapies; challenge of immune evasion

**Table 3 ijms-24-15500-t003:** The multifaceted role of Galectin-1 in PDAC progression and pathways.

Role/Area of Influence	Details/Interactions	Implications for PDAC
Signaling Pathways	ERK, Hedgehog-Gli, RAS, p16INK4a, MAPK, EGFR-Pdx1, TGF-β1/Smad2	Critical in modulation, proliferation, and disease progression
Chemokine Production	Modulates MCP-1, CINC-1 mRNA expression	Enhances inflammation, promoting tumor growth and metastasis
Hh-Gli Pathway Interaction	Regulatory control over genes for migration, adhesion	Vital for PDAC progression and malignancy
RAS Pathway Affinity	Binding to H-Ras and K-Ras	Potential driver in pancreatic cancer progression
p16INK4a Interplay	Influences Gal-1 levels, impacts anoikis	Key player in PDAC cellular dynamics
EGFR-Pdx1 Axis Influence	EGFR-Pdx1 Axis Influence	Direct/indirect impact on PDAC progression
TGF-β1/Smad2 in PSCs	Promotion of feedback loop for fibrosis	Potential target for halting fibrosis in pancreatic cancer
PDAC Cell Migration	Modulation of SDF-1 secretion in PSCs	Possible therapeutic potential in targeting metastasis
EMT Role in PDAC	Inhibition reduces EMT markers, initiates via p38 MAPK pathway	Inhibition reduces EMT markers, initiates via p38 MAPK pathway

Note: PDAC (pancreatic ductal adenocarcinoma), ERK (Extracellular Signal-Regulated Kinase), MCP-1 (Monocyte Chemoattractant Protein-1), CINC-1 (Cytokine-Induced Neutrophil Chemoattractant-1), Hh-Gli (Hedgehog-Gli), EMT (epithelial–mesenchymal transition), PSC (pancreatic stellate cells), MAPK (Mitogen-Activated Protein Kinase), EGFR (Epidermal Growth Factor Receptor), TGF-β1 (Transforming Growth Factor beta 1), Smad2 (SMAD Family Member 2), SDF-1 (Stromal cell-derived factor-1), NF-κB (Nuclear Factor-kappa B).

## Data Availability

As this is a review article, it primarily synthesizes existing research findings rather than presenting new empirical data. All sources and references used for this review are comprehensively cited within the manuscript. Any interested readers or researchers can access the original publications and datasets through the provided references or directly contact the respective authors of the cited works. If there are specific inquiries or need for additional clarifications, please reach out to the corresponding author.

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
