# Peer review of "Galectin-1 in Pancreatic Ductal Adenocarcinoma: Bridging Tumor Biology, Immune Evasion, and Therapeutic Opportunities"

_ijms, 2023, doi:10.3390/ijms242115500_

Round 1

Reviewer 1 Report

The manuscript entitled “Galectin-1 in Pancreatic Ductal Adenocarcinoma: 2 Bridging Tumor Biology, Immune Evasion, and Therapeutic 3 Opportunities ” by Ana Bogut, Bojan Stojanovic, Marina Jovanovic, Milica Dimitrijevic Stojanovic, Nevena Gajovic, Bojana S. Stojanovic, Goran Balovic, Milan Jovanovic, Aleksandar Lazovic, Milos Mirovic, Milena Jurisevic, Ivan Jovanovic and Violeta Mladenovic, provides evidence on the role of Galectin-1 in the pathogenesis of pancreatic ductal adenocarcinoma.

Particularly, this work highlights Galectin-1 pleiotropic functions in influencing tumor growth, since it has recently emerged as a pivotal player in pancreatic cancer microenvironment. The topic examined by the Authors represents a burning issue in the actual literature so it is difficult
to treat in all its facets. Anyway, the argumentation is well exposed. The Authors firstly discuss the stroma complexity, secondly they analyze the structure of Galectin-1 and thirdly they investigate the role of Galectin 1 in tumorigenesis and tumor progression through immune system evasion. In my opinion, the most interesting part is the one which provides therapeutic horizons.

I personally consider the paper acceptable upon revisions.

Specific comments:
1) The Authors should further explain the role of pancreatic stellate cells. It is not clear which is their role as components of the stroma microenvironment and constituents of the extracellular matrix (line 118).
2) The Authors should briefly explain the role of M2 macrophages and the function of T reg lymphocytes since a short comment may enhance the comprehension of the mechanism of tumor immune evasion (line 129).
3) For a more complete argumentation, the Authors should treat the role of immunotherapies in cancer treatment, i.e. when is immunotherapy suggested?
4) The Authors should suggest in which terms a connection between colitis and COVID 19 and pancreatic cancer may be interpreted (line 188).
5) Write from line 190 to line 192 with more fluency.
6) The Authors should provide a link with brain metastases (line 201)
7) It may be interesting that Authors investigate if in IPMN or in PainIN, Gal-1 levels are reduced compared to Gal-1 levels in PDAC.
8) CA 19-9 has been evaluated in screening asymptomatic populations for pancreatic adenocarcinoma in at least two large studies whose conclusion was that screening of asymptomatic subjects using CA 19-9 was ineffective due to the low positive predictive value. Furthermore, screening asymptomatic patients using CA 19-9 may be futile for early detection
of PDAC as CA 19-9 has been shown to be ineffective in small malignant tumors of the pancreas. Nonetheless, CA19-9 has been used to predict tumor stage and resectability, overall survival, and response to therapy in PDAC patients. What may be the function and the efficacy of Gal-1 compared with CA19-9?
9) The Authors should better explain this concept (lines 650-652)
10) The Authors should present future perspective in a less telegraphic way.

Minor comments
1) GRAMMAR AND SPELLING: The Authors should not use contraction (i.e. line 58). It is better to write “to enhance” in line 60.
2) find a synonim of “
foster” (line 185)
3) make a check of acronym and abbreviations. In formal English it is suggested to avoid the use of the Saxon Genitive.

-

Author Response

Dear Reviewer 1,

We extend our heartfelt gratitude for the thorough review and invaluable feedback on our manuscript titled "Galectin-1 in Pancreatic Ductal Adenocarcinoma: Bridging Tumor Biology, Immune Evasion, and Therapeutic Opportunities." Your comments have been vital in directing refinements to enhance the quality of our work.

Specific Comments Response:

Point: The Authors should further explain the role of pancreatic stellate cells. It is not clear which is their role as components of the stroma microenvironment and constituents of the extracellular matrix (line 118).

Response: Recognizing the need for a comprehensive understanding of pancreatic stellate cells, we have expanded the discussion on this subject, delving deeper into their integral role in the stroma microenvironment and as crucial constituents of the extracellular matrix.

Point: The Authors should briefly explain the role of M2 macrophages and the function of T reg lymphocytes since a short comment may enhance the comprehension of the mechanism of tumor immune evasion (line 129).

Response: We've taken heed of your suggestion and added two detailed paragraphs, shedding light on the role and significance of M2 macrophages and Treg lymphocytes in the context of tumor immune evasion.

Point: For a more complete argumentation, the Authors should treat the role of immunotherapies in cancer treatment, i.e. when is immunotherapy suggested?

Response: Acknowledging the importance of this topic, we have crafted an entirely new section dedicated to immunotherapy in PDAC, presenting a comprehensive understanding of its implications, advantages, and challenges.

Point: The Authors should suggest in which terms a connection between colitis and COVID 19 and pancreatic cancer may be interpreted (line 188).

Response: We have accentuated our discussion to better contextualize the relationship between colitis, COVID-19, and pancreatic cancer, with a specific focus on the influential role of Gal-1 across these conditions.

Point: Write from line 190 to line 192 with more fluency.

Response: Based on your feedback, we have restructured lines 190 to 192 to ensure enhanced fluency and coherence.

Point: The Authors should provide a link with brain metastases (line 201).

Response: After thorough research, we found that the literature currently does not provide specific data linking Gal-1 in PDAC to brain metastasis. The term "intralobar" in line 201 specifically refers to pancreatic tissue. We have clarified this in the manuscript to avoid any potential confusion.

Point: It may be interesting that Authors investigate if in IPMN or in PainIN, Gal-1 levels are reduced compared to Gal-1 levels in PDAC.

Response: We've elucidated that while Gal-1 expression is associated with stroma in IPMN and PainIN, in PDAC, there's a pronounced expression in both the stroma and pancreatic carcinomic cells.

Point: CA 19-9 has been evaluated in screening asymptomatic populations for pancreatic adenocarcinoma in at least two large studies whose conclusion was that screening of asymptomatic subjects using CA 19-9 was ineffective due to the low positive predictive value. Furthermore, screening asymptomatic patients using CA 19-9 may be futile for early detection of PDAC as CA 19-9 has been shown to be ineffective in small malignant tumors of the pancreas. Nonetheless, CA19-9 has been used to predict tumor stage and resectability, overall survival, and response to therapy in PDAC patients. What may be the function and the efficacy of Gal-1 compared with CA19-9?

Response: We've expanded on the function and efficacy of Gal-1 in comparison with CA19-9, emphasizing their combined significance and the enhanced potential when used as complementary markers.

Point: The Authors should better explain this concept (lines 650-652).

Response: We have revisited and elaborated upon the concept presented in lines 650-652 to offer greater clarity and understanding.

Point: The Authors should present future perspective in a less telegraphic way.

Response: We've revised the future perspective section, aiming for a more detailed and fluent presentation, ensuring clarity without adopting a telegraphic style.

Minor Comments Response:

Point: GRAMMAR AND SPELLING: The Authors should not use contraction (i.e. line 58). It is better to write “to enhance” in line 60.

Response: We've meticulously addressed the grammar and spelling concerns, ensuring appropriate use of "to enhance".

Point: Find a synonym for “foster” (line 185).

Response: We've replaced the term "foster" with an apt synonym to diversify the language and improve clarity.

Point: Ensure that all terms are defined before the use of their acronyms and ensure consistency in their usage.

Response: We've carefully reviewed the manuscript and ensured that every term is explicitly defined before its acronym is used. Furthermore, we've standardized the use of these acronyms throughout the paper to maintain consistency and clarity.

In conclusion, we genuinely appreciate the detailed feedback and suggestions provided, which have significantly contributed to enhancing the depth, clarity, and overall quality of our manuscript. We are confident that these revisions have addressed all your concerns and hope that the manuscript now aligns more closely with the journal's standards.

Once again, we extend our deepest gratitude for your time and effort in reviewing our work. We look forward to any further feedback you may have.

Warm regards,

Authors

Reviewer 2 Report

This is an excellent and for the most part well written and presented review on the importance of a specific component of the stroma, Galectin-1, in all aspects of PDAC.

There are just some minor problems that I would like to see fixed.

This include a rather stilted English in some parts especially in the Introduction.

Some examples are:

Various risk factors like smoking, obesity, diabetes, and chronic pancreatitis have been linked to PDAC's increasing incidence, emphasizing an urgent need for improved early detection and treatment strategies [6,7]. Risk factors in themselves do not emphasize a need for treatment strategies, this can go after the sentence “Despite surgical interventions and advances in therapeutic approaches, long-term survival (≥ 5 years) remains rare in PDAC, and even very long-term survival (≥ 10 years) is exceptional [8].” OR after:  It's anticipated to become the second leading cause of cancer-related deaths in the United States by 2030 [11].” ALSO, PLEASE FIND AN UPDATED REFERENCE FOR THIS since there are much more recent references than number 11.

Lines 84-87 Through this focused lens on Gal-1 within PDAC, our aim is to enrich the current understanding, ultimately advancing the strategic front against this relentless malignancy. Furthermore, Table 1 provides a structured and concise understanding of the role and impact of Gal-1 in PDAC, summarizing its pivotal interactions and implications in the disease trajectory.

line 95 tumor volume, the stromal components of PDAC, contribute to the unique challenges….

THE Tables are organized such that in Tables 1&2 it is easy to follow the sections but Table 3 is too garbled, there is no clear link between the “Parameter/Aspect” and the “Description/Observation”.

Furthermore, for all the tables: usually there are the references included in the tables for the Observation (right hand columns); if this is too difficult due to space or other reasons then we can let it go.

LASTLY, on Page 20 lines 817-830:

“Inhibitors: Lab trials spotlighted a Galectin-1 inhibitor, LLS2, that augmented the efficacy of chemotherapy agent paclitaxel across multiple human cancer cell lines, encompassing pancreatic cancer cells [148].

Nanotechnology: Nano-oncology is carving out novel avenues to deploy galectin-based inhibitors or galectin ligand-augmented nanoparticles for diverse oncological applications. A recent study pivoted on Gal-1's upregulation in pancreatic cancer for the delivery of magnetic nanoparticles to cancer tissues [148].”

COULD YOU PLEASE INTRODUCE THESE TWO SUBJECTS A BIT BETTER BEFORE THIS SECTION OR FINISHING THE PARAGRAPH JUST BEFORE ; SOMETHING LIKE: More research is imperative to refine the nuances of Gal-1 inhibition, refine inhibitor specificity, and gauge their effectiveness and safety in human trials. ‘IN THIS RESPECT, THERE ARE TWO ASPECTS OF PARTICULAR IMPORTANCE: ‘

WITH THESE SMALL MODIFICATIONS, I AM VERY POSITIVE CONCERNING ITS PUBLICATION

PLEASE SEE MY SUGGESTIONS TO THE AUTHORS ABOVE but I put it also here:

Often there is a rather stilted English in some parts especially in the Introduction.

Some examples are:

Various risk factors like smoking, obesity, diabetes, and chronic pancreatitis have been linked to PDAC's increasing incidence, emphasizing an urgent need for improved early detection and treatment strategies [6,7]. Risk factors in themselves do not emphasize a need for treatment strategies, this can go after the sentence “Despite surgical interventions and advances in therapeutic approaches, long-term survival (≥ 5 years) remains rare in PDAC, and even very long-term survival (≥ 10 years) is exceptional [8].” OR after:  It's anticipated to become the second leading cause of cancer-related deaths in the United States by 2030 [11].” ALSO, PLEASE FIND AN UPDATED REFERENCE FOR THIS since there are much more recent references than number 11.

Lines 84-87 Through this focused lens on Gal-1 within PDAC, our aim is to enrich the current understanding, ultimately advancing the strategic front against this relentless malignancy. Furthermore, Table 1 provides a structured and concise understanding of the role and impact of Gal-1 in PDAC, summarizing its pivotal interactions and implications in the disease trajectory.

line 95 tumor volume, the stromal components of PDAC, contribute to the unique challenges….

Author Response

Dear Reviewer 2,

First and foremost, we would like to express our deepest gratitude for your meticulous review and valuable feedback on our manuscript. Your insights and suggestions significantly contributed to refining and enhancing the quality of our work. Below, we address each of your comments in detail:

Point: Various risk factors like smoking, obesity, diabetes, and chronic pancreatitis have been linked to PDAC's increasing incidence, emphasizing an urgent need for improved early detection and treatment strategies [6,7]. Risk factors in themselves do not emphasize a need for treatment strategies.

Response: We have revised the sentence to read: "The increasing incidence of PDAC has been associated with various risk factors, including smoking, obesity, diabetes, and chronic pancreatitis. This underscores the urgent need for improved early detection and treatment strategies."

Point: It's anticipated to become the second leading cause of cancer-related deaths in the United States by 2030. ALSO, PLEASE FIND AN UPDATED REFERENCE FOR THIS since there are much more recent references than number 11.

Response: We have updated the reference to reflect the most recent data available on this topic.

Point: Lines 84-87 Through this focused lens on Gal-1 within PDAC, our aim is to enrich the current understanding, ultimately advancing the strategic front against this relentless malignancy. Furthermore, Table 1 provides...

Response: We have modified the sentence as per your recommendation for better clarity.

Point: line 95 tumor volume, the stromal components of PDAC, contribute to the unique challenges….

Response: We've adjusted the sentence to: "The stromal components of PDAC, which make up a substantial part of the tumor's size, play a role in the complexities associated with treating this aggressive cancer."

Point: THE Tables are organized such that in Tables 1&2 it is easy to follow the sections but Table 3 is too garbled...

Response: To ensure clarity and coherence, we decided to remove Table 3.

Point: Furthermore, for all the tables: usually there are the references included in the tables for the Observation (right hand columns); if this is too difficult due to space or other reasons then we can let it go.

Response: Due to space constraints, we found it challenging to include the references within the tables. We appreciate your understanding on this matter.

Point: LASTLY, on Page 20 lines 817-830: “Inhibitors: Lab trials spotlighted a Galectin-1 inhibitor, LLS2...

Response: Based on your guidance, we've added an introductory paragraph before discussing inhibitors and nanotechnology: "As we delve deeper into understanding Gal-1's role in PDAC, it becomes increasingly evident that its inhibition offers promising therapeutic potential. To harness this potential fully, further research is imperative. In this context, two avenues are emerging as particularly significant: the development of specific inhibitors and the promising intersection of Gal-1 with nano-oncology."

In conclusion, we deeply appreciate your time and effort in reviewing our manuscript, and we believe that your constructive comments have greatly enhanced its quality. We look forward to any further suggestions or feedback.

With sincere gratitude,

Authors